# Lifelong n-3 Polyunsaturated Fatty Acid Exposure Modulates Size of Mammary Epithelial Cell Populations and Expression of Caveolae Resident Proteins in Fat-1 Mice

**DOI:** 10.3390/nu11102477

**Published:** 2019-10-15

**Authors:** Lyn M. Hillyer, Jing X. Kang, David W.L. Ma

**Affiliations:** 1Department of Human Health and Nutritional Sciences, University of Guelph, Guelph, ON N1G 2W1, Canada; lhillyer@uoguelph.ca; 2Massachusetts General Hospital and Harvard Medical School, Boston, MA 02114, USA; kang.jing@mgh.harvard.edu

**Keywords:** n-3 PUFA, fat-1 mice, caveolae, ER-α, Her-2/neu, breast cancer

## Abstract

Omega-3 polyunsaturated fatty acids (n-3 PUFA) have been associated with reduced breast cancer risk; however, the exact mechanism remains elusive. Female wildtype (WT) and fat-1 mice were fed a 10% safflower diet until 6 weeks of age. Mammary gland epithelial cells (EC) were isolated and EC populations were determined by CD24 surface expression. Fat-1 mice expressed 65%, 20%, and 15% while WT mice expressed 65%, 26% and 9% for non-, myo- and luminal ECs, respectively. The luminal EC population was significantly greater in fat-1 mice (*p* ≤ 0.05), while the total number of mammary ECs were similar between groups (*p* = 0.79). Caveolae was isolated from ECs and Her-2/neu, ER-α and cav-1 protein expression was determined by Western blotting. Fat-1 mice had a two-fold greater ER-α (*p* ≤ 0.05) and a 1.5-fold greater cav-1 (*p* ≤ 0.05) expression than WT with a similar amount of Her-2/neu protein (*p* = 0.990) between groups. Overall, this study provides novel mechanistic evidence by which n-3 PUFA modifies early mammary gland development that may potentially reduce breast cancer risk later in life.

## 1. Introduction

While breast cancer morbidity has declined over the past decade, it remains the most prevalent form of cancer and leading cause of cancer-related mortalities in females worldwide [1,2,3,4,5]. However, incidence rates in Western civilizations are twice as high as those in Asian countries [6,7]. Recent evidence has shown that lifelong consumption of n-3 polyunsaturated fatty acids (PUFA) inhibits mammary tumour growth, as well as modulates mammary gland development [8,9,10]. During puberty female rodents experience expansive terminal end bud (TEB) development which has been identified as the site of mammary tumour initiation [11]. Additionally, the consumption of n-3 PUFA has been shown to reduce the number of TEBs by modulating mammary epithelial cell differentiation and increasing the number of lobular alveolar structures [12].

The TEB within rodent mammary glands are formed from two epithelial cell populations, luminal epithelial cells (CD24^+/high^) and myo-epithelial cells (CD24^+/low^) [13]. Evidence indicates that luminal epithelial cells as the main site for breast cancer initiation while myo-epithelial cells provide ductal integrity and act to suppress tumour growth and invasion [14,15,16]. Additionally, few breast carcinomas contain fully differentiated myo-epithelial cells and the ones which do, lack the ability to produce sufficient amounts of functional laminin-1, a key regulator of luminal epithelial cell polarity [15,16,17]. The lack of mature myo-epithelial cells in mammary gland tumours may indicate a loss of ductal structure, as well as a reduction in cell-mediated signaling with the luminal epithelia.

Caveolae are plasma membrane domains found in abundance on the surface of epithelial cells and act as key regulatory sites for signal transduction. Due to their fluidic nature, caveolae are thought to play a role in cellular diseases, such as breast cancer [18,19]. Three major proteins associated with caveolae and breast cancer are caveolin-1 (cav-1), human epidermal growth factor receptor 2 (Her-2/neu) and estrogen receptor-α (ER-α). Cav-1 is the principle quantitative marker for caveolae and is believed to have tumour-suppressing capabilities by improving caveolae formation and consequently, cell signaling [19,20,21]. The over expression of ER-α and Her-2/neu are biomarkers for aggressive types of breast cancer as activation of either receptor stimulates proliferatory pathways by increasing DNA replication, which increases the possibility of cancer-causing mutations [22,23].

In this preliminary study, the transgenic fat-1 mouse was used to determine the effects of n-3 PUFA on mammary gland epithelial cell populations and signalling molecules found within caveolae. The fat-1 mouse can endogenously synthesize n-3 PUFA from n-6 PUFA precursors via an omega-3-desaturase transgene (C. elegans) found in all tissues, thus, avoiding any diet-associated complications that can arise with rodent feeding trials [24]. The objective of this study was two-fold, (1) to determine the effect of n-3 PUFA on mammary gland development, specifically by altering epithelial cell populations (non-epithelial (CD24^-^), myo-epithelial (CD24^+/low^) and luminal epithelial (CD24^+/high^) cells); and (2) to determine cav-1, ER-α and Her-2/neu expression in caveolae extracted from mammary glands of the fat-1 mouse.

## 2. Materials and Methods

### 2.1. Ethics Statement

This investigation was performed under Animal Utilization Protocol 07G010, which was approved by the Animal Care Committee of the University of Guelph under the governance of the Canadian Council on Animal Care. Each mammary gland sample was pooled from 6–8 mice. Thus, animal numbers were minimized to the full extent permitted by statistical rigour. For this reason, *n* = 3 for both the WT and fat-1 groups.

### 2.2. Animals, Diets and Phenotyping

Fat-1 mice, originally obtained from Dr. Jing Kang (Harvard Medical School), were used from an in-house breeding colony and husbandry practices are reported elsewhere [10]. Harems were fed a modified AIN93G diet (Research Diets Inc.) ad libitum containing 10% fat (*w*/*w*) from safflower oil; providing 22% of the mouse’s total daily energy requirements (Table 1). Offspring were weaned at 3 weeks of age, and phenotyped as described previously [10]. Female offspring were maintained on their parental diet for 6–8 weeks until termination.

### 2.3. Euthanization and Tissue Collection

On the day of termination, the stage of the mouse’s estrus cycle was determined by a vaginal smear. Mice in diestrus were maintained until they passed into proestrus to control for hormonal fluctuations which may impact cell proliferation profiles [25]. Mice in other stages of the estrus cycle were terminated by CO_2_ overdose and the right and left 4th and 5th mammary glands (MG) were excised for epithelial cell isolation.

### 2.4. Epithelial Cell Characterization and Caveolae Isolation

The 4th and 5th MGs were pooled from 6–8 mice per group (*n* = 3); lymph nodes were removed, and mammary epithelial cells were isolated using the method of Prater et al. [26]. Briefly, finely minced MGs were digested in collagenase/hyaluronidase (StemCell, cat # 07912) for 18 h at 37 °C to allow for tissue dissociation. Cells were washed in Hank’s balanced salt solution (Sigma, cat # H6648) and treated with ammonium chloride (Sigma, cat # A9434), Trypsin/EDTA (Sigma, cat # T4049), dispase (StemCell, cat # 07913) and DNAase 1 (Sigma, cat # D5025) to release the epithelial cells.

An aliquot (2.5 × 10^5^) of isolated epithelial cells was used to determine CD24 surface expression by flow cytometry. Cells were stained on ice for 30 min with 0.5 µg/µL FITC-conjugated CD24 (ebiosciences, cat # 11-0242-82, rat IgG2b, clone: M1/69), washed with 2 mL, 1% BSA/PBS and stained on ice for a further 30 min with 0.2 µg/µL PE-conjugated CD45 (ebiosciences, cat # 12-0451-82, rat IgG2b, clone 30-F11). Cells were washed with 2 mL 1% BSA/PBS and resuspended in 100 µL 1% BSA/PBS for flow cytometry using a Becton Dickenson FACSCalibur E4272 flow cytometer. Each analysis included an isotype control, namely FITC-conjugated Rat IgG2b (ebiosciences, cat # 11-4031-81) for CD24 and PE-conjugated Rat IgG2b (ebiosciences, Cat # 12-4031-81) for CD45. For flow cytometer analysis, cells were gated on the CD45PE-negative population and epithelial populations were defined based on the intensity of CD24FITC expression. Non-epithelial cells are CD24^-^, myo-epithelial cells are CD24^+/low^ and luminal epithelial cells are CD24^+/high^. Each analysis was comprised of 10^4^ viable cells.

Caveolae were isolated from the remaining MG epithelial cells as described by Macdonald and Pike [27]. After the last ultracentrifugation step, eleven–1 mL fractions were collected starting from the liquid surface. Fractions 5–8 were caveolae-rich and were pooled for further analysis.

### 2.5. Protein Expression

Protein levels of enriched caveolae were determined by the Bradford protein assay (BioRad, Cat # 5000001). Quantities of 1, 2 and 5 µg of protein were ran through 15% acrylamide gel with a 5% stacking gel for cav-1, ER-α and Her-2/neu, respectively, to determine protein expression by Western protein blotting. The separated proteins were transferred to a polyvinylidene fluoride (PVDF) membrane, blocked overnight in 5% skim milk powder and incubated for 2 h at room temperature with either caveolin-1 antibody (Santa Cruz, Cat # sc-894) diluted 1:500 or ER-α antibody (Santa Cruz, Cat # sc-7207) diluted 1:200 or Her2 neu antibody (Santa Cruz, Cat # sc-101695) diluted 1:200. All primary antibodies were followed by goat anti-rabbit IgG HRP (Santa Cruz, Cat # sc-2030) diluted 1:1000 for 1 h at RT. Protein bands were detected by Western Lightening Plus ECL (Perkin Elmer, Cat # NEL 103001EA) and visualized on a FluorChem HD2 imager (Cell Biosciences, Santa Clara, California, USA). Bands were quantified using AlphaView imaging software (Version 3.1.1.0).

### 2.6. Phospholipid Fatty Acid Analysis

Lipids were extracted from isolated caveolae via the Folch Method [28] and phospholipid fractions were separated by thin layer chromatography (TLC) [29], as previously described. Briefly, H-plates (EMD Chemicals #5721-7) were incubated in an oven for one hour at 100 °C prior to being spotted by samples. They were then placed in a TLC tank containing 30 mL chloroform, 9 mL methanol, 25 mL 2-propanol, 6 mL 0.25 M KCl, and 18 mL triethylamine. Following this, the plates were then lightly sprayed with 0.1% (*w*/*v*) ANSA (Fluka #GA 12046) before being visualized under UV light. Bands corresponding to phosphatidylcholine (PC) and phosphatidylethanolamine (PE) were collected.

Methylation was performed by adding 2 mL of hexane and 2 mL of 14% BF_3_-MeOH (Sigma B1252) to the samples and incubating them at 100 °C for 90 min. Following methylation, 2 mL of double-distilled H_2_O was added to the samples and the solution was immediately vortexed for 30 s to halt methylation. Samples were centrifuged for 10 min at 357× *g*, the hexane layer was collected and dried down under nitrogen before reconstitution in 50 µL of hexane. Fatty acid methyl esters were quantified on an Agilent 6890 gas chromatograph equipped with flame ionization detection and separated on a DB-FFAP fused-silica capillary column (15 m, 0.1 m film thickness, 0.1 mm i.d.; Agilent Cat # 127-32H2). Samples were injected in 200:1 split mode. The injector and detector ports were set at 250 °C. Fatty acid methyl esters were eluted using a temperature program set initially at 150 °C and held for 0.25 min, increased at 35 °C/min and held at 170 °C for 3 min, increased at 9 °C/min to 225 °C, and finally, increased 80 °C/min to 245 °C and held for 2.2 min. The run time per sample is 12 min. The carrier gas was hydrogen, set to a 30 mL/min constant flow rate. Peaks were identified by retention times of fatty acid methyl ester standards (Nu-Chek-Prep, Elysian, MN) using EZchrom Elite version 3.2.1 software. Fatty acid results were calculated as percent composition.

### 2.7. Statistical Analysis

Data was subjected to a two-tailed Student’s *t*-test with a pre-set upper limit of probability for statistical significance set at *p* = 0.05. SAS system for windows, version 8.2, was used for analyses.

## 3. Results

### 3.1. Epithelial Cell Populations

Based on CD24 expression, the fat-1 mouse had a significantly altered MG epithelial cell profile, as shown in the flow cytometer histograms (Figure 1A,B) and the percent of epithelial cells expressing CD24 (Figure 2).

The total number of mammary epithelial cells obtained from the fat-1 and WT mice were not different (*p* = 0.79) (Figure 3A). The fat-1 mouse expressed 65% non-epithelial, 20% myo-epithelial and 15% luminal epithelial cells compared to the WT mouse, which expressed 65% non-epithelial, 26% myo-epithelial, and 9% luminal epithelial cells (Figure 2). The luminal epithelial cell profile was significantly greater in the fat-1 mouse (*p* ≤ 0.05) (Figure 1 and Figure 2).

### 3.2. Protein Expression

The fat-1 mouse acquired an altered expression of cav-1 and ER-α proteins in comparison to the WT mouse. In contrast to WT mice, the fat-1 mice had a 1.5-fold increase in cav-1 (*p* ≤ 0.05) expression (Figure 3B); a two-fold increase in ER-α (*p* ≤ 0.05) expression (Figure 3D); while the amount of Her-2/neu protein was not significantly different (*p* = 0.990) between groups (Figure 3C).

### 3.3. Phospholipid Fatty Acids

Within the PC fraction of the MG epithelial cells’ caveolae, the fat-1 mice had significant increases in α-linolenic acid, eicosapentaenoic acid and docosahexaenoic acid (*p* ≤ 0.05), and complementary decreases in linoleic acid and arachidonic acid (*p* ≤ 0.05) (Table 2). A similar trend was observed within the PE fraction of the same caveolae (Table 2).

## 4. Discussion

Our lab has previously shown that lifelong exposure to n-3 PUFAs reduces both tumour volume and multiplicity in the mammary glands of MMTV-neu(ndl)-YD5 mice [8,9]. While it has been proposed that n-3 PUFA have the ability to alter MG development, the mechanism of action remains poorly understood. The current study has shown that the endogenous production of n-3 PUFA from an n-6 PUFA source has the capacity to significantly increase the luminal epithelial cell population while not altering the total number or size of epithelial cells found within the mouse’s MG. Luminal epithelial cells exhibit stem cell characteristics and have the ability to differentiate into myo-epithelial cells [13]. Breast cancer is believed to arise in the luminal epithelial compartment but the relationship between luminal and myo-epithelial lineages require further examination [13]. An increase in luminal epithelial cell abundance within the MG may indicate a greater risk towards developing MG tumours. However, a greater marker of breast cancer risk than elevated amounts of luminal epithelial cells is a complete loss of myo-epithelial cells [15,17]. While the percentage of myo-epithelial cells within the fat-1 transgenic mice mammary glands was non-significantly reduced compared to wildtype mice (20% compared to 26%, respectively), they still had a large proportion of myo-epithelial cells, which may provide the required ductal integrity and cellular signals to prevent tumour development.

Further, n-3 PUFA have also been shown to alter the expression of tumour-promoting and tumour-suppressing proteins. Cav-1 is a constituent protein and principle marker of caveolae. In addition, the over-expression of cav-1 has been associated with tumour suppression [21], while ER-α and Her-2/neu are tumour promoters [22,23]. The observed increase in cav-1 and ER-α may not be contradictory given that normal cells were assessed. Thus, in the context of mammary gland development, the increase of cav-1 expressed in the fat-1 mouse may have protective effects against mammary tumour growth later in life. Also, recent evidence shows that cav-1 overexpression is associated with increased ER-α in caveolae and this protein–protein interaction inhibits ER-α signaling [30]. Furthermore, while the increased expression of ER-α may indicate a greater risk of developing cancer, it must be noted that tumours expressing increased ER-α lead to a promising prognosis for cancer treatment [22]. ER-α-positive cancers are often successfully treated by endocrine therapy, whereas ER-α-negative tumours have a poor prognosis and, generally, cannot be treated [22]. Whether this increases the risk of breast cancer or improves the prognosis of breast cancer, should it develop, requires further research with a mammary tumour mouse model. Her-2/neu is over-expressed in breast cancer and is a specific tumour marker [23]. There was no change in the expression levels of Her-2/neu between the fat-1 and WT mice. This is likely reflective of the absence of tumours in these otherwise healthy mice. While a focus of this study has been at the level of caveolae, downstream signalling, total cellular protein expression of these proteins and gene expression were not assessed. These are important measures for future investigation in order to obtain a complete understanding of molecular, cellular and functional changes.

In conclusion, these results suggest that n-3 PUFA produced endogenously within the fat-1 mouse can alter the proportion of epithelial populations in the developing pubertal mammary gland, which may have implications for future cancer risk. Additionally, n-3 PUFA may protect against long-term development of breast cancer by influencing cell signaling pathways mediated by caveolae during pubertal MG development at 6–8 weeks of age. Further research is warranted to conclude whether the epithelial cell profile and cav-1 expression of the fat-1 mouse is protective against breast cancer.

## Figures and Tables

**Figure 1 nutrients-11-02477-f001:**
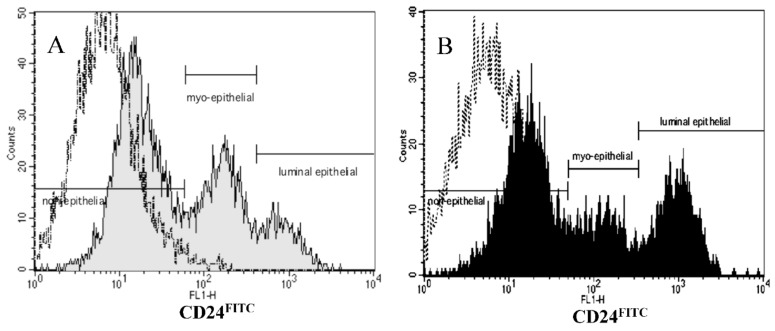
Representative flow cytometer histograms of mammary epithelial cell preparations for wildtype (**A**; grey) and fat-1 (**B**; black) mice. (**A**,**B**), epithelial cell populations determined by CD24^FITC^ staining intensity after gating out CD45PE^+^ cells. Rat IgG2b^FITC^ isotype control shown in black dotted lines.

**Figure 2 nutrients-11-02477-f002:**
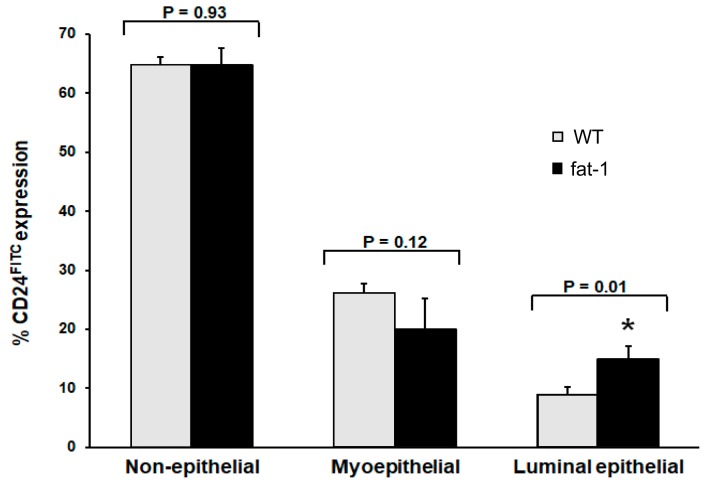
Percent CD24^FITC^ surface marker expression on mammary epithelial cells. Bars represent means plus SD and * identifies differences (*p* ≤ 0.05) between wildtype (WT) and fat-1 mice by Student’s *t*-test.

**Figure 3 nutrients-11-02477-f003:**
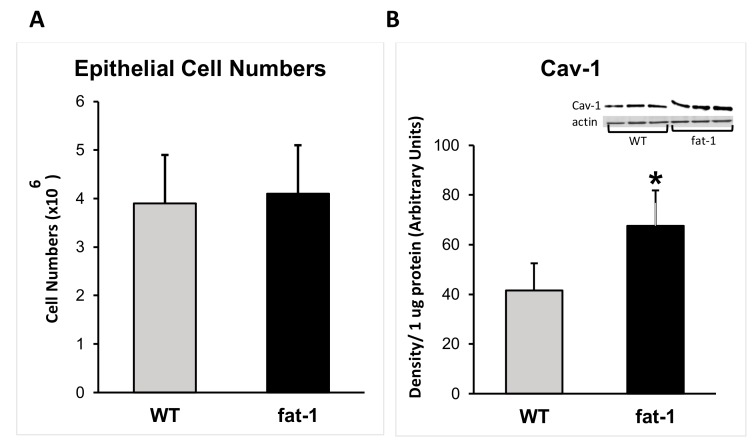
Epithelial cell numbers and protein expression, including representative Western blots, in caveolae isolated from mammary epithelial cells of wildtype (WT) and fat-1 mice: (**A**) epithelial cell numbers, (**B**) Cav-1, (**C**) Her-2/neu, (**D**) ER-α. * denotes a significant difference at *p* ≤ 0.05 by Student’s *t*-test.

**Table 1 nutrients-11-02477-t001:** Composition of 10% safflower diet fed to wildtype (WT) and fat-1 mice.

Ingredient	g/Kg
Casein	200
L-cystine	3.0
Cornstarch	336.7
Maltodextrin	132
Sucrose	100
Cellulose	50
Safflower oil	97
t-butylhydroquinone	0.019
Mineral mix S10022G	35
Vitamin Mix V10037	10
Choline bitartrate	2.5

**Table 2 nutrients-11-02477-t002:** Percent fatty acid composition of caveolae isolated from mammary epithelial cells of wildtype (WT) and fat-1 mice: PC and PE phospholipid fractions ^1^.

	PC			PE		
	WT	Fat-1	*p* Value	WT	Fat-1	*p* Value
18:0, stearic	44.1 *	34.6	≤0.05	50.9	47.1	0.116
18:1, oleic	19.4	16.7	0.371	29.2	32.7	0.214
18:2n6, linoleic	24.0 *	15.2	≤0.05	11.3 *	4.7	≤0.05
18:3n3, alpha linolenic	0.2	3.9 *	≤0.05	0	1.4 *	≤0.05
20:4n6, arachidonic	12.3 *	8.6	≤0.05	8.6 *	2.6	≤0.05
20:5n3, eicosapentaenoic	0	8.8 *	≤0.05	0	5.6 *	≤0.05
22:6n3, docosahexaenoic	0	12.1 *	≤0.05	0	5.9 *	≤0.05

^1^ Means of selected fatty acids are reported. Within a row and phospholipid fraction, values with a * differ (*p* ≤ 0.05) according to Student’s *t*-test between WT and fat-1 mice. PC: phosphatidylcholine, PE: phosphatidylethanolamine.

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
