# Peer review of "Lifelong n-3 Polyunsaturated Fatty Acid Exposure Modulates Size of Mammary Epithelial Cell Populations and Expression of Caveolae Resident Proteins in Fat-1 Mice"

_nutrients, 2019, doi:10.3390/nu11102477_

Round 1

Reviewer 1 Report

This paper entitled “Lifelong n-3 Polyunsaturated Fatty Acid Exposure 2 Modulates Size of Mammary Epithelial Cell Populations and Expression of Caveolae Resident Proteins in Fat-1 Mice” is excellently written and drives the point that the research results support that n-3 polyunsaturated fatty acids produced endogenously within the fat-1 transgenic mice appears capable of altering the proportion of epithelial populations in developing pubertal mammary gland. That this model may serve usefully in future cancer risk, is presented.   What makes this paper particularly interesting is that it employs fat-1 transgenic mice which are capable of producing n-3 fatty acids from the n-6 type, leading to abundant n-3 fatty acids with reduced levels of n-6 fatty acids in their organs and tissues. Because of this, these transgenic mice have been employed in many studies that focus on the effects of n-3 fatty acids on numerous health problems.  As well noted within this manuscript, the use of these transgenic mice ideally avoid any diet associated complications that could be expected with rodent feeding trials.  This is significant study.

The introduction is well written with an excellent discussion of caveolae and of their importance within the plasma membrane domain. The materials and methods section is also very well-written and clearly presents detailed information required for this manuscript.

In agreement with this manuscripts conclusions, this investigation provides novel mechanistic evidence by which n-3 polyunsaturated fatty acids modify mammary gland in their early development that may potentially reduce breast cancer risk later in life.

Reviewer 2 Report

The manuscript entitled “Lifelong n-3 polyunsaturated fatty acid exposure modulates size of mammary epithelial cell populations and expression of caveolae resident proteins in fat-1 mice” by LM Hillyer et al., (nutrients-584499) reports the effects of n-3 polyunsaturated fatty acids (PUFA) over both the percentage of epìthelial cell populations in mammary glands and the expression of several proteins found in caveolae of mammary gland cells. These studies are conducted both in wild type and in fat-1 transgenic mice fed lifelong with a diet rich in n-6 PUFA, which are transformed to n-3 PUFA in transgenic mice. The authors also characterize the fatty acid composition of epithelial cell caveolae in these mice, founding a significant increase in n-3 PUFA and a concomitant decrease in n-6 PUFA in fat-1 mice.

Some issues that the authors should address are the following.

Broad comments:

- My main concern when reviewing this manuscript is related to the data included in Figure 3, and the biological significance of the changes shown in the quantification of band intensity in Western blots. In my opinion, the interpretation of this Figure by the readers would be straighter if the western blots (or at least representative examples of them) are also shown in the Figure. Are these differences kept when analysing complete cells (i.e.: are these differences due to differential expression of Cav-1 and ER-α in fat-1 and WT mammary gland cells, or alternatively are these proteins equally expressed -but differentially distributed in caveolae-?). Due to the importance of these results in the manuscript, I would suggest to include the images of Western blots; to discuss in more depth if the differences found are restricted to caveolae or affect the whole mammary gland; or even to confirm by other means (as RT-qPCR or Western blot of complete mammary glands) the differences found in Cav-1 and ER-α abundance in caveolae of fat-1 and WT mice.

Specific comments:

-I wonder if the authors have studied the histology of mammary glands of fat-1 and WT mice fed with 10% safflower diet. Does the increased percentage of luminal (CD24high) epithelial cells found in fat-1 mice affects the histology of mammary glands? If the authors have carried out this study, I strongly suggest its inclusion it in this manuscript.

- fat-1 is sometimes in italics and sometimes not. Please, homogenize.

- Table 2. Interpretation of this Table would be easier if the names of the fatty acids are included in a new column. In this Table, an asterisk is missing in the percent composition of 22:6n3 in the phosphatidylcholine fraction of fat-1 mice.

- For homogeneity with Figures 2 and 3, I suggest to represent the flow cytometry histogram of WT mice shown in Figure 1 in black, and that of fat-1 mice in grey or white.

- In lanes 69-70, it is not clear in which experiments n=3 and in which ones the experimental glands were pooled from 6-8 mice. Equally, it is not clear which are the 3 groups mentioned in lane 86.

- Lane 76: There are no details of how to genotype mice in ref 10. Another reference is needed.

- In Material and Methods (lines 109-117), the method used to quantify the data obtained in Western blots should be included.

- Line 114. ER-α. The Greek letter α is missing.

- Lines 148-150 (and Figure 1). It is not clear which are the panels 1A, 1B, 2A and 2B in Figure 1. It seems that two of the panels of Figure 1 are missing.

. Line 181 and Figure 2. In the comparison of luminal epithelial cells between fat-1 and WT mice it is not clear the actual p-value (p=0.01 In Figure 2; p=0.005 in line 181).

-Line 183 (legend to Figure 3). Epithelial cell numbers has nothing to do with caveolae. In my opinion, the legend should to be changed to: “Epithelial cell numbers and protein expression in caveolae isolated from mammary epithelial cells of WT and fat-1 mice: A) …”

- Lines 185 and 197. There is the same typo in both lines (an extra dot at the end of the sentence in line 195, and between “of” and “WT” in line 197.

Author Response

Reviewer 2 Comments and Responses

Comments and Suggestions for Authors

The manuscript entitled “Lifelong n-3 polyunsaturated fatty acid exposure modulates size of mammary epithelial cell populations and expression of caveolae resident proteins in fat-1 mice” by LM Hillyer et al., (nutrients-584499) reports the effects of n-3 polyunsaturated fatty acids (PUFA) over both the percentage of epìthelial cell populations in mammary glands and the expression of several proteins found in caveolae of mammary gland cells. These studies are conducted both in wild type and in fat-1 transgenic mice fed lifelong with a diet rich in n-6 PUFA, which are transformed to n-3 PUFA in transgenic mice. The authors also characterize the fatty acid composition of epithelial cell caveolae in these mice, founding a significant increase in n-3 PUFA and a concomitant decrease in n-6 PUFA in fat-1 mice.

Some issues that the authors should address are the following.

Broad comments:

- My main concern when reviewing this manuscript is related to the data included in Figure 3, and the biological significance of the changes shown in the quantification of band intensity in Western blots. In my opinion, the interpretation of this Figure by the readers would be straighter if the western blots (or at least representative examples of them) are also shown in the Figure.

Response: We have endeavoured to improve Figure 3 as suggested. Specifically, the representative bands have been included.

Are these differences kept when analysing complete cells (i.e.: are these differences due to differential expression of Cav-1 and ER-α in fat-1 and WT mammary gland cells, or alternatively are these proteins equally expressed -but differentially distributed in caveolae-?). Due to the importance of these results in the manuscript, I would suggest to include the images of Western blots; to discuss in more depth if the differences found are restricted to caveolae or affect the whole mammary gland; or even to confirm by other means (as RT-qPCR or Western blot of complete mammary glands) the differences found in Cav-1 and ER-α abundance in caveolae of fat-1 and WT mice.

Response: We have not analyzed total cell lysates because the goal of this initial paper was to examine if changes were evident in the caveolae fraction.  We agree with this limitation and have edited the text in the discussion as follows,

lns 280-283. “While a focus of this study has been at the level of caveolae, downstream signalling, total cellular protein expression of these proteins and gene expression were not assessed. These are important measures for future investigation in order to obtain a complete understanding of molecular, cellular and functional changes.”

Specific comments:

-I wonder if the authors have studied the histology of mammary glands of fat-1 and WT mice fed with 10% safflower diet. Does the increased percentage of luminal (CD24high) epithelial cells found in fat-1 mice affects the histology of mammary glands? If the authors have carried out this study, I strongly suggest its inclusion it in this manuscript.

Response: We have not examined the corresponding histology of the mammary gland in tandem with changes in CD24high.  We agree that this would be an important measure and have plans to examine this relationship in the future research.

- fat-1 is sometimes in italics and sometimes not. Please, homogenize.

Response: fat-1 without italics has been changed throughout the manuscript

- Table 2. Interpretation of this Table would be easier if the names of the fatty acids are included in a new column. In this Table, an asterisk is missing in the percent composition of 22:6n3 in the phosphatidylcholine fraction of fat-1 mice.

Response: The names of fatty acids have been added to the table and the missing asterisk has been added.

- For homogeneity with Figures 2 and 3, I suggest to represent the flow cytometry histogram of WT mice shown in Figure 1 in black, and that of fat-1 mice in grey or white.

Response: Due to software limitations we were not able to alter figure 1. Figures 2 and 3 have been changed to match figure 1.

- In lanes 69-70, it is not clear in which experiments n=3 and in which ones the experimental glands were pooled from 6-8 mice. Equally, it is not clear which are the 3 groups mentioned in lane 86.

Response: All experiments came from the same sample size of 3.  However, each sample was pooled from 6-8 mice.  The text has been revised to make this clearer. The revised text in lines 69-80 now states, “Each mammary gland sample was pooled from 6-8 mice. Thus, animal numbers were minimized to the full extent permitted by statistical rigour. For this reason, n = 3 for both the WT and fat-1 groups.”

- Lane 76: There are no details of how to genotype mice in ref 10. Another reference is needed.

Response: We apologize for the confusion. Genotyping has been removed from the manuscript. These mice were only phenotyped by gas chromatography and confirmed by their lipid profile, which we do routinely.

- In Material and Methods (lines 109-117), the method used to quantify the data obtained in Western blots should be included.

Response: The Western blot method has been included in brief as suggested, ln 126.

- Line 114. ER-α. The Greek letter α is missing.

Response – alpha has been added.

- Lines 148-150 (and Figure 1). It is not clear which are the panels 1A, 1B, 2A and 2B in Figure 1. It seems that two of the panels of Figure 1 are missing.

Response:  The text has been corrected to show A and B only.

. Line 181 and Figure 2. In the comparison of luminal epithelial cells between fat-1 and WT mice it is not clear the actual p-value (p=0.01 In Figure 2; p=0.005 in line 181).

Response: All significant p values have been changed to p ≤ 0.05 throughout the manuscript for consistency.

-Line 183 (legend to Figure 3). Epithelial cell numbers has nothing to do with caveolae. In my opinion, the legend should to be changed to: “Epithelial cell numbers and protein expression in caveolae isolated from mammary epithelial cells of WT and fat-1 mice: A) …”

Response: The figure legend has been changed as suggested by the reviewer.

- Lines 185 and 197. There is the same typo in both lines (an extra dot at the end of the sentence in line 195, and between “of” and “WT” in line 197.

Response:  The typo has been corrected.

Round 2

Reviewer 2 Report

All the issues included in my review have been adequately answered.